# Elevated plasma cotinine is associated with an increased risk of developing IBD, especially among users of combusted tobacco

**Lovisa Widbom**[1]*, **Jörn Schneede**[2], **Øivind Midttun**[3], **Per Magne Ueland**[4], **Pontus Karling**[5], **Johan Hultdin**[1]

1 Department of Medical Biosciences, Clinical Chemistry, Umeå University, Umeå, Sweden, 2 Department of Pharmacology and Clinical Neuroscience, Clinical Pharmacology, Umeå University, Umeå, Sweden, 3 Bevital AS, Bergen, Norway, 4 Medicine and Pathology, Haukeland University Hospital, Bergen, Norway, 5 Department of Public Health and Clinical Medicine, Medicine, Umeå University, Umeå, Sweden

* lovisa.widbom@umu.se

## Abstract

### Objective

Smoking has previously been associated with inflammatory bowel disease (IBD), but no study has reported on cotinine, an objective, biochemical measure of tobacco use. We aimed at testing the hypothesis that cotinine levels among healthy subjects are associated with an increased risk of developing IBD in later life.

### Design

We analysed plasma cotinine and evaluated corresponding lifestyle questionnaires that included tobacco habits in subjects (n = 96) who later developed late-onset IBD (70 ulcerative colitis (UC) and 26 Crohn's disease (CD)) and in sex and age-matched controls (n = 191).

### Results

Patients who later developed IBD had significantly higher plasma cotinine levels compared to controls. In multivariable analysis, higher log-cotinine was associated with a higher risk of developing IBD (OR 1.34 (95% CI 1.01–1.63)). After stratifying for time to diagnosis, the association was only significant in subjects with shorter time (< 5.1 years) to diagnosis (OR 1.45 (1.09–1.92)). The findings were similar for UC- and CD-cases, but did not reach statistical significance in CD-cases. Although plasma cotinine concentrations were higher in snuff users compared to combusted tobacco users, no increase in the risk of IBD and lower risk of developing IBD among subjects with shorter time (< 5.1 years) to diagnosis was seen among snuff users.

### Conclusions

Cotinine, a biomarker of tobacco use, is associated with increased risk of developing late-onset IBD in general, and UC in particular. No increased risk among snuff users indicates

**Data Availability Statement:** Aggregate data are included in the manuscript and its Supporting Information files. Individual data is not publicly available, as it contains potentially identifying

patient information. Data are available upon request from the Northern Sweden Health and Disease Study Biobank. To request data, interested researchers must complete a formal application (available at https://www.umu.se/en/biobank-research-unit/research/access-to-samples-and-data/access-to-nsdd/) and submit it to The Biobank Research Unit at Umeå University (contact via ebf@umu.se).

**Funding:** This work was supported by the Västerbotten County Council [grant numbers VLL-678111 and VLL-582981]. Biobank Sweden was supported by the Swedish Research Council [grant number VR 2017-00650]. Bevital AS provided support in the form of salary for author ØM but did not have any additional role in the study design, data collection and analysis, decision to publish, or preparation of the manuscript. The specific roles of this author are articulated in the 'author contributions' section.

**Competing interests:** Author ØM was employed by Bevital AS, this does not alter our adherence to PLOS ONE policies on sharing data and materials. No other competing interests declared.

that other components in combusted tobacco than nicotine may be involved in the pathogenesis of IBD among smokers.

## Introduction

Tobacco is a major risk factor of a wide spectrum of diseases and in general the greatest preventable cause of premature death. Still, in the 21st century, a significant proportion of the world's population is consuming tobacco. In Sweden, approximately 10.7% of inhabitants smoke daily, i.e. use combusted tobacco (2016–2017) [1].

Smoking has also been associated with the risk of inflammatory bowel disease (IBD) [2]. A recent systematic review concludes that the incidence of IBD is decreasing in countries where the smoking prevalence is decreasing, indicating a connection. The same study also reported that the proportion of never smokers developing IBD is increasing, indicating that smoking is not the only risk factor [3].

Smoking is the only proven environmental determinant for Crohn's disease (CD) [4], and smokers with CD often have a more complicated course of disease than non-smokers [5].

The association between smoking and ulcerative colitis (UC) is possibly even more complex. Current smokers who stop smoking appear to be more prone to develop the disease, but there is no clear beneficial effect of tobacco smoking on the natural course of UC [6]. Recently, we showed that "never smokers" exhibit a lower risk for late-onset UC [7]. Specific components in smoked/combusted tobacco that may deteriorate IBD have not been identified so far. The same is true for possible protective factors of smoking that may modify the disease activity in patients with UC [8].

Smoking is associated with inflammatory activity. For example, a study on 1819 participants from the PLCO Cancer Screening Trial showed an association between biochemical markers of inflammation and smoking [9]. One study on peripheral blood mononuclear cells from 37 IBD patients and 17 controls showed that cytokine profiles differ in IBD patients compared to healthy controls and that these cytokine profiles may be modified by nicotine [10].

Cotinine is a metabolite of nicotine, with a longer half-life, which makes it suitable as a marker for long-term tobacco use [11, 12]. Cotinine concentrations in urine, saliva, serum and plasma are all associated with smoking intensity [13, 14]. Cotinine is considered a better marker for quantifying smoking than self-reporting of smoking [15].

Nicotine and cotinine have previously been analysed in one study on 51 UC patients who also were smokers [16]. Reported median serum concentrations of nicotine and cotinine were 8 ng/mL (49.3 nmol/L) and 180 ng/mL (1021.5 nmol/L), respectively. An increase in nicotine of 12.1 ng/mL (74.12 nmol/L) 2 minutes after smoking one cigarette was observed. While there are prospective studies on the associations between smoking risk and IBD [17], no study has assessed cotinine as a measure of smoking exposure in relation to IBD-risk.

In Sweden, the use of moist, smokeless, non-combusted tobacco (snuff) is common, with 12.9% of inhabitants using snuff daily (2016–2017) [1]. Smokeless tobacco, such as snuff, has been associated with higher levels of cotinine compared to smoking [18], Therefore, we have in this study chosen to analyse snuff use as well as smoking. Snuff use has not been associated with health risks in the same way as smoking has, and there was no association between snuff use and the development of IBD in a study on 1100 UC and 700 CD patients among Swedish construction workers [17].

The present study aimed to investigate whether pre-diagnostic cotinine levels predict the development of IBD in later life and to compare the predictive value of cotinine with self-reported smoking exposure and snuff use.

## Materials and methods

### Ethical approval

The regional ethical board, Umeå, Sweden (Dnr 06-024M, 2010-284-31 M), approved the study. Written informed consent was obtained from all participants before inclusion in the study.

### Study population

Study participants were selected from patients with known IBD, confirmed by objective measures, i.e. colonoscopy, MRI or biopsy. All participants were recruited from the Västerbotten intervention project (VIP) or the Mammography screening project (MA), both within the Northern Sweden Health and Disease Study (NSHDS).

The VIP started in 1985 and is a health screening intervention project, which annually invites all inhabitants of the Västerbotten County in northern Sweden turning 40, 50 and 60 years to a health screening at their primary care centre. At these screenings, survey data and blood samples are collected and stored in a biobank that presently contains data from over 150 000 visits. The MA project was active between the years 1995–2006. Blood samples and survey data were collected in conjunction with mammography screening for women aged 18–82 years. The biobank contains about 54 000 samples.

Initially, all participants from the VIP or MA, with an ICD-10-diagnosis of IBD (K50.1–9 or K51.1–9) were included. All diagnoses were reaffirmed by reviewing medical records, and the participants with an incorrect diagnosis were excluded.

The time of IBD diagnosis was defined as the first objective finding (i.e. MRI, colonoscopy or biopsy) based on reviews of the patient's medical records. Subjects diagnosed with IBD less than one year after participating in the NSHDS health examination were excluded from the study. Information about what type of IBD, UC or CD, was also collected from the medical records.

For each case, two controls, matched for age, gender and time and centre of health screening, were randomly selected among healthy participants from the NSHDS cohort. A total of 96 patients (70 UC and 26 CD) eligible for the study and 191 matched controls were available to be included in further analyses.

### Lifestyle data

The participants smoking status and use of smokeless tobacco were obtained from questionnaire data.

The questionnaires were filled in, and the blood samples were collected at the same health screening. Length and weight were also measured during the visit. The questionnaire contained information on previous and current tobacco use, including type, amount (cigarettes per day for smokers and snuff boxes per week for snuff users), duration of use and time since quitting.

Participants smoking or using snuff when completing the questionnaire were defined as active users. Previous smokers or snuff users were categorized as non-smokers or non-snuff users. From the medical records the data on the present use of tobacco at the time of diagnosis was not accurate enough to be included in the study.

### Sample collection and handling

Blood samplings were performed after overnight fasting, after resting for 15 minutes venous blood was collected in 10 mL Na-heparin tubes, centrifuged at 1500 G for 15 minutes, and aliquots of plasma were frozen within one hour and stored at -80˚C.

## Biochemical analyses

Plasma cotinine was analysed with liquid chromatography-tandem mass spectrometry (LC-MS/MS) as reported previously [19]. The lower limit of detection was 1.0 nmol/L, and within-day and between-day coefficients of variation were 2–3% and 6%, respectively.

## Statistical analyses

All statistical analyses were made using IBM SPSS statistics version 24.0 (IBM Corporation, New York, NY, USA). Differences between continuous variables were assessed using the Mann-Whitney U-test, and for categorical variables, the Chi$^2$-test was used. Calculations were performed with IBD as outcome in the total study group and after stratification for the type of IBD (UC or CD).

Subjects with cotinine levels <85 nmol/L were defined as non-tobacco user and those with cotinine ≥85 nmol/L as tobacco users [20]. These subgroups were further divided into four categories; subjects with cotinine levels <5 nmol/L was defined as non- tobacco users, 5–85 nmol/L as passive tobacco users, >85–1700 nmol/L as tobacco users and >1700 nmol/L as heavy tobacco users [20].

For conditional logistic regression, log-transformed cotinine values were used as the distribution is highly skewed. Both univariable and multivariable conditional logistic regression were performed to assess the odds ratio (OR) for developing IBD. The multivariable model included log-cotinine, smoking and snuff use. For both univariable and multivariable models, the subjects were further stratified into two sub-groups based on the median time to diagnosis. When calculating p for trend, variables were treated as continuous variables in regression analysis.

P-values <0.05 were considered statistically significant.

## Results

### Baseline characteristics

Baseline characteristics are shown in Table 1. No differences between cases and controls were seen for age, gender, BMI or snuff use. Self-reported smoking was more common, and plasma cotinine concentrations were higher among those who developed IBD in later life compared to matched controls. For smoking, we observed similar results for UC and CD; however, cotinine was higher in UC, but not in CD-patients as compared to controls (S1 Table). In the CD group, self-reported number of cigarettes per day was higher compared to matched controls; no differences were seen for UC or all IBD subjects.

Using cotinine to define tobacco users /non-tobacco users showed a higher frequency of (cotinine-defined) tobacco users in the IBD-group compared to matched controls. This was also seen in the UC-, but not in the CD-group. Further sub-categorization based on cotinine for differentiation between passive and heavy tobacco users showed a significant difference between cases and controls in the IBD group as a whole, but this was no longer significant for UC or CD subjects.

Cotinine levels for different types of tobacco exposure are shown in Table 1, separately for cases and controls. There were no differences in cotinine levels between cases and controls for smokers, snuff users or users of both snuff and cigarettes (Table 1). Cotinine levels among non-users were higher among cases compared to controls (Table 1), also seen for UC, but not for CD cases (S1 Table).

There was a significant correlation between the self-reported number of cigarettes/day and categories of cotinine in plasma (Spearman correlation coefficient 0.62, p<0.001).

**Table 1. Baseline characteristics for subjects who later developed IBD (case) and controls.**

| | Case | Control | p-value[*] | n Case/Control |
|---|---|---|---|---|
| Age, years | 50.1 (40.1–59.8) | 50.0 (40.1–59.7) | 0.80 | 96/191 |
| Time to diagnosis, years | 5.09 (2.66–7.23) | n.a. | n.a. | 96/n.a. |
| Gender, women (%) | 50.0 | 53.4 | 0.68 | 96/191 |
| BMI, kg/m$^2$ | 25.1 (23.2–28.5) | 25.4 (23.1–28.0) | 0.83 | 87/172 |
| Smoking (%) | 31.3 | 19.4 | **0.041** | 87/170 |
| Number of cigarettes/day (%) | | | | 83/164 |
| 0 | 61.5 | 70.7 | | 59/135 |
| 1–4 | 4.2 | 2.6 | | 4/5 |
| 5–14 | 13.5 | 9.9 | | 13/19 |
| 15–25 | 7.3 | 2.6 | 0.15 | 7/5 |
| Snuff use (%) | 16.7 | 14.7 | 0.97 | 87/162 |
| Cotinine, nmol/L | 5.96 (1.32–1350) | 1.45 (<1.00–917) | **0.001** | 95/190 |
| Cotinine categories (%) | | | | 95/190 |
| Non-tobacco users | 52.1 | 69.6 | | 50/133 |
| Tobacco users | 46.9 | 29.8 | **0.006** | 45/57 |
| Cotinine subcategories (%) | | | | 95/190 |
| Non-tobacco users | 49.0 | 63.4 | | 47/121 |
| Passive-tobacco users | 3.1 | 6.3 | | 3/12 |
| Tobacco users | 32.3 | 20.4 | | 31/39 |
| Heavy tobacco users | 14.6 | 9.4 | **0.033** | 14/18 |
| Cotinine, nmol/L (by type of tobacco) | | | | 86/160 |
| Smokers | 1060 (614.0–1425) | 1185 (617.0–1495) | 0.58 | 24/24 |
| Snuff users | 1670 (1370–2750) | 1360 (1238–2080) | 0.19 | 11/18 |
| Smoking and snuff-use | 2120 (1305–2400) | 1770 (1108–2308) | 0.68 | 5/10 |
| Non-users | 1.56 (0.75–2.22) | 1.05 (0.00–1.88) | **0.010** | 46/108 |

Values are median (25–75 percentiles) for continuous variables, and proportions for categorical variables.

[*]Calculated with Mann-Whitney for continuous variables and Chi$^2$ for categorical variables.

Cotinine categories–Non-tobacco users: <85 nmol/L, tobacco users: ≥85 nmol/L. Cotinine subcategories–Non-tobacco users: <5 nmol/L, passive tobacco users: 5-<85 nmol/L, tobacco users: 85–1700 nmol/L, heavy tobacco users: >1700 nmol/L.

n.a.: Not applicable.

Plasma cotinine concentrations were higher in snuff users compared to smokers, median (25th-75th percentile) 1490 (1270–2230) vs 1145 (641–1435) nmol/L, p <0.001.

### Univariable conditional logistic regression

There was a significant trend for developing IBD with a higher number of self-reported cigarettes per day (p = 0.023), but there was no significance in the risk for IBD when analysing each category of number of cigarettes (Table 2). After subdividing into UC and CD, this was only seen in the CD group (S2 Table).

Cotinine-defined tobacco users showed a higher OR for developing IBD compared to controls (OR 2.00; 95% CI 1.21–3.28), and further dividing into cotinine-based sub-categories showed the same results (Table 2). When stratifying for IBD subtype, the same was seen for UC (OR 2.21; 95% CI 1.21–4.06), but not for CD cases (S2 Table).

A higher log-Cotinine, analysed as a continuous variable, was associated with a higher OR for developing IBD (OR 1.24; 95% CI 1.11–1.40), when stratifying for the median time from

**Table 2. Conditional logistic regression, univariable odds ratios (OR) and 95% confidence intervals (CI) and p for trend across categories, for developing inflammatory bowel disease and categories of tobacco exposure.**

| Tobacco exposure | OR (95%CI) | p trend | n Case/Control |
|---|---|---|---|
| Number of cigarettes/day | | | 83/156 |
| 0 | Ref | | |
| 1–4 | 1.37 (0.36–5.26) | | |
| 5–14 | 1.89 (0.86–4.18) | | |
| 15–25 | 3.18 (0.97–10.4) | **0.023** | |
| Cotinine categories | | | 95/188 |
| Non-tobacco users | Ref | | |
| Tobacco users | **2.00 (1.21–3.28)** | 0.006 | |
| Cotinine subcategories | | | 95/188 |
| Non-tobacco users | Ref | | |
| Passive-tobacco users | 0.70 (0.17–2.84) | | |
| Tobacco users | **1.97 (1.10–3.50)** | | |
| Heavy tobacco users | 1.82 (0.83–4.00) | **0.013** | |

Cotinine categories–Non-tobacco users: <85 nmol/L, tobacco users: ≥85 nmol/L. Cotinine subcategories–Non-tobacco users: <5 nmol/L, passive tobacco users: 5-<85 nmol/L, tobacco users: 85–1700 nmol/L, heavy tobacco users: >1700 nmol/L.

blood-sampling to diagnosis, the same was seen for both groups (Table 3). When stratifying for IBD subtype, the same was seen for UC (OR 1.33; 95% CI 1.14–1.55), but not for CD cases (S3 Table).

Self-reported smoking was associated with a higher OR for developing IBD (OR 1.90; 95% CI 1.06–3.42), although after stratifying by median time from sampling to diagnosis (5.09 years), this was no longer significant (Table 3). No association between self-reported smoking and risk for developing the disease was seen after subdividing cases in a UC and a CD group (S3 Table).

**Table 3. Conditional logistic regression, univariable odds ratios (OR) and 95% confidence intervals (CI) for developing inflammatory bowel disease, subdivided by median time from data collection to diagnosis.**

| Tobacco exposure | OR (95%CI) | n Case/Control |
|---|---|---|
| All cases | | |
| log-Cotinine | **1.24 (1.11–1.40)** | 95/188 |
| Smoking | **1.90 (1.06–3.42)** | 87/170 |
| Snuff use | 1.02 (0.49–2.16) | 87/162 |
| Data collected <5 years before diagnosis* | | |
| log-Cotinine | **1.24 (1.05–1.45)** | 48/96 |
| Smoking | 1.59 (0.62–4.06) | 44/87 |
| Snuff use | 0.78 (0.29–2.11) | 44/84 |
| Data collected >5 years before diagnosis* | | |
| log-Cotinine | **1.25 (1.06–1.48)** | 47/92 |
| Smoking | 2.13 (<1.00[†]-4.55) | 43/83 |
| Snuff use | 1.47 (0.48–4.46) | 43/78 |

*5.09 years, the median time before diagnosis.

[†] 0.999975.

**Table 4. Multivariable conditional logistic regression showing odds ratios (OR) and confidence interval (CI) for inflammatory bowel disease, subdivided by median time from data collection to diagnosis.**

|  | OR (95% CI) | n Case/Control |
|---|---|---|
| All cases |  | 86/158 |
| log-Cotinine | **1.34 (1.10–1.63)** |  |
| Smoking | 0.88 (0.39–1.97) |  |
| Snuff use | 0.40 (0.15–1.02) |  |
| Data collected <5 years before diagnosis* |  | 44/83 |
| log-Cotinine | **1.45 (1.09–1.92)** |  |
| Smoking | 0.42 (0.11–1.69) |  |
| Snuff use | **0.23 (0.06–0.95)** |  |
| Data collected >5 years before diagnosis* |  | 42/75 |
| log-Cotinine | 1.30 (0.96–1.75) |  |
| Smoking | 1.21 (0.43–3.47) |  |
| Snuff use | 0.58 (0.15–2.31) |  |

Three variables were included in all multivariable models: log-Cotinine, smoking and snuff use.

*5.09 years, the median time before diagnosis.

No associations were seen between snuff use and risk for IBD, not in the whole group, nor any of the sub-groups (Table 3; S3 Table).

## Multivariable conditional logistic regression

In a multivariable model including log-Cotinine, smoking and snuff use, higher log-Cotinine was associated with a higher OR for developing IBD when analysing the whole group (OR 1.34; 95% CI 1.10–1.63) (Table 4). When sub-dividing by median time from blood-sampling to diagnosis, this was also seen for log-Cotinine in the group with data collected less than 5.09 years before diagnosis (OR 1.45 95% CI 1.09–1.92). In the same sub-group (samples collected less than 5.09 years before diagnosis), snuff use was associated with a lower risk for developing IBD (OR 0.23; 95% CI 0.06–0.95) (Table 4).

In the UC sub-group, higher cotinine was associated with a higher risk of developing the disease (OR 1.42; 95% CI 1.12–1.79). When stratifying for the median time from blood-sampling to diagnosis, this was only seen in the group with data collected less than 5.09 before diagnosis (OR 1.78; 95% CI 1.12–2.81) (S4 Table).

No associations were seen between tobacco exposure and risk for CD in the multivariable models (S4 Table).

## Discussion

We found that cotinine levels were associated with increased risk of developing IBD, and UC in particular. Smoking has earlier been linked to IBD and is an established risk factor for CD, but its role in UC appears to be more complex. Up until now, all evidence of associations between smoking and IBD have been based on self-reported exposure. Self-reported data are prone to bias and objectifiable; quantitative data of smoking exposure in relation to IBD-risk would be desirable. This is the first study to report the potential impact of tobacco exposure as measured by plasma cotinine concentrations on the risk of developing IBD. Cotinine has earlier been demonstrated being an accurate estimate of nicotine exposure [15]. In this prospective cohort study, cotinine in plasma was associated with an increased risk of developing IBD

in later life. Similar associations were found when using predefined [20] cotinine cut-offs for defining tobacco use. In addition to IBD, this association was also statistically significant in UC-cases, both before and after sub-grouping according to median time between blood sampling and IBD-diagnosis. Thus, our data confirm findings from earlier studies on associations of self-reported smoking with IBD.

The findings are in concordance with our previous study, showing for the first time that smoking status, based on questionnaires with self-reported smoking, was associated with a higher risk of developing UC, CD and IBD [7]. In the present study, these findings were confirmed and extended for cotinine, smoking and snuff use in multivariable analysis. Cotinine was associated with IBD risk in a model that differentiated between combusted and non-combusted tobacco use. In addition, the proportion of snuff users was lower among subjects developing IBD within five years from inclusion. The OR remained low in all models (although insignificant), which may indicate an elevated risk from combusted tobacco but not from non-combusted tobacco. This may also imply that nicotine (measured as cotinine) per se may not be a causal factor, but rather a proxy for other risk factors from combusted tobacco.

Many different pathophysiological factors associated with smoking, including vitamins, immunity, inflammation, antioxidants, vascular factors, gut permeability and motility [21], as well as effects on the gut microbiota [22] have been implicated in IBD.

The effects of nicotine in IBD is unclear. Nicotine has been shown to induce Neutrophil extracellular traps (NETs) in in vitro studies, which may be involved in inflammatory activity in IBD [23]. Based on biopsies, NETs were present in virtually all patients with IBD [24, 25]. In contrast, nicotine may protect against colitis by mechanisms involving micro RNA-124 and STAT3 in mice [26].

In humans there appear to be a risk of rebound effect upon smoking cessation, resulting in an increased risk of developing UC [8]. Changes in cytokines in IBD patients that smoke have been reported, i.e. IL-8 (inducing chemotaxis of granulocytes) being lower in smokers compared to non-smokers with IBD with contradictory findings among controls [27]. Nicotine may also have an effect on apoptosis and cell cycle regulation [10].

Cotinine was higher among snuff users compared to smokers in our study. This is in line with a previous study suggesting that the first-pass metabolism in the liver of swallowed nicotine from chewing tobacco and snuff results in higher plasma cotinine compared to nicotine from smoking [28]. This could explain the higher cotinine seen among snuff users compared to smokers. Still, there is a difference in the association of combusted and non-combusted tobacco with IBD risk in the present study, indicating that factors other than nicotine from combusted tobacco might contribute to the risk of IBD. The route of administration of nicotine or the tissue primarily affected may determine the type of immune response and thereby influence disease development.

There are some limitations to the present study. Firstly, this study mainly addresses patients with late-onset IBD due to the design of NSHDS. Late-onset IBD represents only about one-quarter of all IBD cases (23%) [29]. Therefore, the effect of cotinine in this study may not necessarily apply to younger patients. Secondly, not all participants reported their tobacco use; therefore, in analyses on self-reported data, the number of subjects was somewhat lower compared to subjects with plasma analyses. Thirdly, the data on the medical records on tobacco use was not accurate enough to be included in the analysis. Finally, the sub-group that later developed CD was small in number, which might explain the lack of a statistically significant association between cotinine and the development of CD.

The strength of the present study is its prospective nature; all data were collected years before diagnosis, reducing the risk for recall bias. The cases were thoroughly ascertained and matched with two controls. Blood samples were handled according to standardized pre-

defined protocols, and cotinine in plasma was analysed with a gold standard LC-MS/MS method with high analytical performance. We were also able to separately study the effects of both combusted and non-combusted tobacco.

## Conclusion

Exposure to cigarette smoking as determined by cotinine in plasma is a risk factor for late-onset IBD, including UC. The possible lower risk for IBD in users of non-combusted tobacco (snuff) in a multivariable model indicates that other components than nicotine may be involved in the pathogenesis of IBD in smokers.

## Supporting information

**S1 Table. Baseline characteristics for subjects who later developed ulcerative colitis and Crohn's disease (cases) and matched controls.** Median (25–75 percentile) for continuous variables, proportions for categorical variables.
(DOCX)

**S2 Table. Conditional logistic regression, univariable odds ratios (OR) and 95% confidence intervals (CI) and p for trend across categories, for developing ulcerative colitis and Crohn's disease, and categories of tobacco exposure.**
(DOCX)

**S3 Table. Conditional logistic regression, univariable odds ratios (OR) and 95% confidence intervals (CI) for developing ulcerative colitis and Crohn's disease, subdivided by median time from data collection to diagnosis.**
(DOCX)

**S4 Table. Multivariable conditional logistic regression showing odds ratios (OR) and confidence interval (CI) for ulcerative colitis and Crohn's disease, subdivided by median time from data collection to diagnosis.**
(DOCX)

## Acknowledgments

We thank the participants of the NSHDS for their contribution to this study. We thank the Department of Biobank Research, at Umeå University, the Västerbotten Intervention Project and Västerbotten County Council for providing data and samples and acknowledge the contribution from Biobank Sweden.

## Author Contributions

**Conceptualization:** Lovisa Widbom, Pontus Karling, Johan Hultdin.

**Data curation:** Lovisa Widbom, Pontus Karling, Johan Hultdin.

**Funding acquisition:** Pontus Karling, Johan Hultdin.

**Methodology:** Jörn Schneede, Øivind Midttun, Per Magne Ueland, Pontus Karling, Johan Hultdin.

**Project administration:** Pontus Karling.

**Resources:** Jörn Schneede, Øivind Midttun, Per Magne Ueland, Pontus Karling, Johan Hultdin.

**Writing – original draft:** Lovisa Widbom, Johan Hultdin.

**Writing – review & editing:** Jörn Schneede, Øivind Midttun, Per Magne Ueland, Pontus Karling.

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
