## [Decision Letter · Decision Letter 0]

13 Jan 2020

PONE-D-19-31583

Elevated plasma cotinine is associated with an increased risk of developing IBD, especially among users of combusted tobacco

PLOS ONE

Dear Dr. Lovisa Widbom,

Thank you for submitting your manuscript to PLOS ONE. After careful consideration, we feel that it has merit but does not fully meet PLOS ONE’s publication criteria as it currently stands. Therefore, we invite you to submit a revised version of the manuscript that addresses the points raised during the review process.

We would appreciate receiving your revised manuscript by April 30, 2020. To enhance the reproducibility of your results, we recommend that if applicable you deposit your laboratory protocols in protocols.io, where a protocol can be assigned its own identifier (DOI) such that it can be cited independently in the future. For instructions see: http://journals.plos.org/plosone/s/submission-guidelines#loc-laboratory-protocols

We look forward to receiving your revised manuscript.

Kind regards,

Stefanos Bonovas, M.D., M.Sc., Ph.D.

Academic Editor

PLOS ONE

Journal Requirements:

2. Thank you for including your ethics statement in your manuscript. We ask that you please additionally include that you obtained written informed consent from the participants of this study in your statement as well

'The authors have declared that no competing interests exist.'

We note that one or more of the authors are employed by a commercial company: Bevital AS.

Additional Editor Comments (if provided):

Reviewers' comments:

Reviewer's Responses to Questions

**Comments to the Author**

1. Is the manuscript technically sound, and do the data support the conclusions?

Reviewer #1: Partly

Reviewer #2: Partly

2. Has the statistical analysis been performed appropriately and rigorously? 

Reviewer #1: No

Reviewer #2: Yes

3. Have the authors made all data underlying the findings in their manuscript fully available?

Reviewer #1: Yes

Reviewer #2: Yes

4. Is the manuscript presented in an intelligible fashion and written in standard English?

Reviewer #1: No

Reviewer #2: Yes

5. Review Comments to the Author

Reviewer #1: I found this very interesting. The novel finding is that cotinine levels do not predict the risk of IBD developing in snuff users (higher levels of cotinine than smokers but not higher levels of IBD) suggesting some component of cigarette smoke other than nicotine is the causative factor increasing the risk of IBD.

However the manuscript does need significant revision and I do not think it can be accepted in its current form. The results section is very unclear and difficult to read. The authors describe their results in the text but do not put the actual statistics in the text. Rather they require the reader to look at "Table 2" to look up the relevant statistic. I think it would be better to follow the convention of having the statistic in brackets e.g. (OR x.xx, 95% CI x.xx-x.xx, Table 2).

I also find some of the results section misleading. For example "A higher number of self-reported cigarettes per day was associated with higher OR for developing IBD (Table 2)." When looking at table two I see that none of the ORs are statistically significant, all of the 95% CIs cross 1.0 but instead the authors are reporting a "p trend" on the values of the odds ratios themselves in relation to the number of cigarettes smoked. This is not described in the methods section and it should be clear exactly which statistical test was used.

My other major concern is that in supplementary table 2 it suggests the OR for developing Crohn's in a heavy smoker is 148631 relative to a non-smoker. This is clearly incorrect and likely results from the study being very under-powered. While I am not a statistician by training my experience is that such outlandish results are almost always incorrect and the statistical analysis has been performed incorrectly.

I also note that they have pre-defined cut-off levels of cotinine to determine smoking status. How did these concur with the patients' self reported smoking? Also when reading reference 20 I couldn't see these cut-off levels used but perhaps I missed this.

Finally the English, while intelligible and certainly better than my Swedish, is non-standard. In the first paragraph the authors refer to the 21th rather than the 21st century. They also refer to "medical journals" which to me would mean publications such as PLOS ONE but I believe they mean "medical records". I think it would be important to have a native English speaker to proof read the manuscript before re-submission.

Reviewer #2: Thank you for the opportunity to review the manusrcipt entitled "Elevated plasma cotinine is associated with an increased risk of developing IBD, especially among users of combusted tobacco". The topic is highly interesting but the manuscript would need a bit of a revision.

The Introduction is brief and clear.

In the Materials and Methods it should be added that the matching was not complete in a 1:2 maner.

Why were not previous smokers/snuff users as previous users rather than non-users? This may affect the findings.

Also, and even more important, there is a continuous unclearity throughout the manuscript (and tables) on cotinine levels being the cut off for smokers. Should this not be changed to tobacco users instead as smokers and snuff users later are analysed separately? Causes confusion in it's present form.

In the Results I would prefer to have cotinine levels evaluated first and then separate for smokers and snuff users.

Regarding all tables I would like to have them moved from a separate column at the end to be added in each column respectively. Further you should add % in the columns as this is unclear at the moment.

I do not get the numbers in the tables to add up as expected. E.g. in table 1 Smoking (tobacco user?) is 87/170 but regarding # of cigarettes/day 83/164. In table 2 # cig/day is only 83/156. Non-smokers in table 1 is 96/161 while in table 2 85/188. Same disparities apply for tables 3 and 4.

Was there any data on smoking status at time of diagnosis? Would be highly interesting, especially regarding the increased risk for UC patients.

In the discussion there I would like more discussion on the found risk for UC among smokers/tobacco users. Also there is a recent Swedish publication for incidence for late onset of IBD (Everhov et al) which might be more appropriate in a national epidemiological maner.

6. PLOS authors have the option to publish the peer review history of their article (what does this mean?). If published, this will include your full peer review and any attached files.

Reviewer #1: No

Reviewer #2: No

---

## [Author Response · Author response to Decision Letter 0]

20 May 2020

To enhance the reproducibility of your results, we recommend that if applicable you deposit your laboratory protocols in protocols.io, where a protocol can be assigned its own identifier (DOI) such that it can be cited independently in the future. 

Response: The method has been published previously, ref 19 in the manuscript: Midttun Ø, Hustad S, Ueland PM. Quantitative profiling of biomarkers related to B-vitamin status, tryptophan metabolism and inflammation in human plasma by liquid chromatography/tandem mass spectrometry. Rapid Commun Mass Spectrom 2009;23:1371-9.

Journal requirements 

 Response: Done.

2. Thank you for including your ethics statement in your manuscript. We ask that you please additionally include that you obtained written informed consent from the participants of this study in your statement as well

Response: Added under ethical approval: Written informed consent was obtained from all participants before inclusion in the study.

3. We note that you have indicated that data from this study are available upon request. PLOS only allows data to be available upon request if there are legal or ethical restrictions on sharing data publicly. 

 Response: All relevant data are within the paper and its Supporting Information files.

Individual data is not provided as it can be linked to individuals, in order to access individual data, a formal application to the northern Sweden health and disease study biobank can be made.

 4. Please include captions for your Supporting Information files at the end of your manuscript, and update any in-text citations to match accordingly.

Response: Supporting information captions added to the manuscript and in-text citations updated.

5. We note that one or more of the authors are employed by a commercial company: Bevital AS.

* Please provide an amended Funding Statement declaring this commercial affiliation, as well as a statement regarding the Role of Funders in your study. If the funding organization did not play a role

in the study design, data collection and analysis, decision to publish, or preparation of the manuscript and only provided financial support in the form of authors' salaries and/or research materials,please review your statements relating to the author contributions, and ensure you have specifically and accurately indicated the role(s) that these authors had in your study. You can update author roles in

the Author Contributions section of the online submission form.

“The funder provided support in the form of salaries for authors [insert relevant initials], but did not have any additional role in the study design, data collection and analysis, decision to publish,

or preparation of the manuscript. The specific roles of these authors are articulated in the ‘author contributions’ section.”

Please also provide an updated Competing Interests Statement declaring this commercial affiliation along with any other relevant declarations relating to employment, consultancy, patents, products in

development, or marketed products, etc.

Within your Competing Interests Statement, please confirm that this commercial affiliation does not alter your adherence to all PLOS ONE policies on sharing data and materials by including the following

statement: "This does not alter our adherence to PLOS ONE policies on sharing data and materials.”.

If this adherence statement is not accurate and there are restrictions on sharing of data and/or materials, please state these. Please note that we cannot proceed with consideration of your article until this nformation has been declared.

Please include both an updated Funding Statement and Competing nterests Statement in your cover letter. We will change the online

submission form on your behalf.

Response: The text “Bevital AS provided support in the form of salary for author ØM, but did not have any additional role in the study design, data collection and analysis, decision to publish, or preparation of the manuscript. The specific roles of this author are articulated in the 'author contributions' section.” has been added under funding.

The text “Author ØM was employed by Bevital AS: This does not alter our adherence to PLOS ONE policies on sharing data and materials. No other competing interests declared.” has been added under competing interests. 

REVIEWER 1 

The results section is very unclear and difficult to read. The authors describe their results

in the text but do not put the actual statistics in the text. Rather they require the reader to look at "Table 2" to look up the relevant statistic. I think it would be better to follow the convention of having the statistic in brackets e.g. (OR x.xx, 95% CI x.xx-x.xx,Table 2).

Response: For relevant significant results we have included p-values, OR and 95% CI in the text as suggested.

I also find some of the results section misleading. For example "A higher number of self-reported cigarettes per day was associated with higher OR for developing IBD (Table 2)." When looking at table two I see that none of the ORs are statistically significant, all of the 95%

CIs cross 1.0 but instead the authors are reporting a "p trend" on the values of the odds ratios themselves in relation to the number of

cigarettes smoked. This is not described in the methods section and it should be clear exactly which statistical test was used.

Response: We have included a clarification that there is a significant trend for IBD with increasing numbers of cigarettes. We have in the method section explained p for trend.

My other major concern is that in supplementary table 2 it suggests the OR for developing Crohn's in a heavy smoker is 148631 relative to

a non-smoker. This is clearly incorrect and likely results from the study being very under-powered. While I am not a statistician by

training my experience is that such outlandish results are almost always incorrect and the statistical analysis has been performed

incorrectly.

Response: We agree that the number 148631 is an obscure number, resulting from very few heavy smokers (n=4) and no controls. It is evident that this is not significant by looking at the confidence intervals. No conclusions can be drawn from this but we have included this to present all data of the study. 

I also note that they have pre-defined cut-off levels of cotinine to determine smoking status. How did these concur with the patients' self reported smoking? Also when reading reference 20 I couldn't see these cut-off levels used but perhaps I missed this.

Response: The reason for the values being different in the reference is because of use of different units, the reference using ng/mL, whereas we have used mmol/L.

There was a significant Spearman correlation between self-reported number of cigarettes/day and categories of cotinine in plasma (p<0.001) in the total material of this study, although in some individuals the cotinine concentration does not correspond to number of cigarettes reported. The cotinine concentrations are a more accurate measure of smoking compared to self-reported smoking data, reducing bias from reporting. Added to results.

Finally the English, while intelligible and certainly better than my Swedish, is non-standard. In the first paragraph the authors refer to the 21th rather than the 21st century. They also refer to "medical journals" which to me would mean publications such as PLOS ONE but I believe they mean "medical records". I think it would be important to have a native English speaker to proof read the manuscript before re-submission.

 Response: The English language has been corrected.

REVIEWER 2 

In the Materials and Methods it should be added that the matching was not complete in a 1:2 maner.

Response: One control was lost due lack of subjects at one center. In the method section we added the sentence “..191 matched controls were available to be included in further analyses.”

Why were not previous smokers/snuff users as previous users rather than non-users? This may affect the findings.

Response: There were 67 subjects who were reported as previous smokers and 33 subjects who were reported previous snuff users. The questionnaires did not include information at what time stopped using tobacco neither for how longtime they used tobacco or degree of smoking. Therefore, we believe that these subjects were better handled in the group “non-smokers”. This was a conservative approach as this could reduce difference between groups. Despite this classification we showed a clear increased risk for smokers to develop IBD.

Also, and even more important, there is a continuous unclearity throughout the manuscript (and tables) on cotinine levels being the

cut off for smokers. Should this not be changed to tobacco users instead as smokers and snuff users later are analysed separately?

Response: Thank you for the suggestion, we have changed to tobacco users for categories defined by cotinine.

In the Results I would prefer to have cotinine levels evaluated first and then separate for smokers and snuff users.

Response: In table 1 we added cotinine-levels for smokers, snuff-users, users of both snuff and cigarettes and for those not using any tobacco. 

Regarding all tables I would like to have them moved from a separate column at the end to be added in each column respectively. Further you

should add % in the columns as this is unclear at the moment.

Response: We have added ”%” in the tables. However, due to different missing subject in each category we think that a separate column is necessary to state number of subject in each analysis (see below).

I do not get the numbers in the tables to add up as expected. E.g. in table 1 Smoking (tobacco user?) is 87/170 but regarding # of

cigarettes/day 83/164. In table 2 # cig/day is only 83/156. Non-smokers in table 1 is 96/161 while in table 2 85/188. Same disparities apply for tables 3 and 4.

Response: The study includes different parameters for defining smoking. Unfortunately, there was no complete data for all the included parameters due to missing values in the questionnaires and one patient and one control subject was lost in the cotinine analysis.

1. Number of cigarettes: Based on questionnaires (n=83/166)

2. Smokers and non-smokers: Based on questionnaires (n=87/170)

3. Smokers defined by Cotinine measurements (n=95/190)

In the multivariate analysis the number drops.

Was there any data on smoking status at time of diagnosis? Would be highly interesting, especially regarding the increased risk for UC patients.

Response: We agree, but unfortunately, the data on tobacco use at the time of diagnosis of IBD was not accurate enough to be included in the study.

We have added a comment on this both in the method section and in the discussion section.

In the discussion there I would like more discussion on the found risk for UC among smokers/tobacco users. Also there is a recent Swedish publication for incidence for late onset of IBD (Everhov et al) whichmight be more appropriate in a national epidemiological maner.

 Response: In the discussion section we have added Everhov et al as a reference.

---

## [Decision Letter · Decision Letter 1]

18 Jun 2020

Elevated plasma cotinine is associated with an increased risk of developing IBD, especially among users of combusted tobacco

PONE-D-19-31583R1

Dear Dr. Widbom,

We’re pleased to inform you that your manuscript has been judged scientifically suitable for publication and will be formally accepted for publication once it meets all outstanding technical requirements.

Kind regards,

Stefanos Bonovas, M.D., M.Sc., Ph.D.

Academic Editor

PLOS ONE

Additional Editor Comments (optional):

Reviewers' comments:

Reviewer's Responses to Questions

**Comments to the Author**

1. If the authors have adequately addressed your comments raised in a previous round of review and you feel that this manuscript is now acceptable for publication, you may indicate that here to bypass the “Comments to the Author” section, enter your conflict of interest statement in the “Confidential to Editor” section, and submit your "Accept" recommendation.

Reviewer #2: All comments have been addressed

Reviewer #3: All comments have been addressed

2. Is the manuscript technically sound, and do the data support the conclusions?

Reviewer #2: Yes

Reviewer #3: Yes

3. Has the statistical analysis been performed appropriately and rigorously? 

Reviewer #2: Yes

Reviewer #3: Yes

4. Have the authors made all data underlying the findings in their manuscript fully available?

Reviewer #2: Yes

Reviewer #3: Yes

5. Is the manuscript presented in an intelligible fashion and written in standard English?

Reviewer #2: Yes

Reviewer #3: Yes

6. Review Comments to the Author

Reviewer #2: I find that the authors have commented to all questions and changed the manuscript accordingly and improving the manuscript satisfactory.

Reviewer #3: Authors regurarly responded to all the questions, and the suggestions, posed by the previous reviewers. No major issues arised from this second revision.

7. PLOS authors have the option to publish the peer review history of their article (what does this mean?). If published, this will include your full peer review and any attached files.

Reviewer #2: No

Reviewer #3: No